# Knowledge, Attitudes, and Practices towards PrEP from Cisgender Men and Transgender Women Who Have Sex with Men in the Largest Suburban HIV Epidemic

**DOI:** 10.3390/ijerph191811640

**Published:** 2022-09-15

**Authors:** Andrew Spieldenner, Anthony J. Santella, Spring C. Cooper, Kathleen Rosales, William C. Goedel, Heidi E. Jones

**Affiliations:** 1Department of Communication and Media Studies, California State University, San Marcos, CA 92096, USA; 2Marion Peckham Egan School of Nursing and Health Studies, Fairfield University, Fairfield, CT 06824, USA; 3Department of Community and Social Sciences, City University of New York School of Public Health, New York, NY 10027, USA; 4Department of Epidemiology and Biostatistics, City University of New York School of Public Health, New York, NY 10027, USA; 5Department of Epidemiology, Brown University School of Public Health, Providence, RI 02903, USA

**Keywords:** HIV/AIDS, PrEP, gay and bisexual men, suburban health

## Abstract

We conducted a cross-sectional web-based study to assess attitudes and experiences with HIV pre-exposure prophylaxis (PrEP) amongst a multiracial cohort of gay, bisexual, and other men who have sex with men (MSM) and transgender women in Long Island, New York. Participants were recruited through clinical providers and community-based organizations. The survey assessed knowledge and attitudes toward PrEP and factors that facilitate willingness to take it. Of the 189 respondents, most participants were Latinx/Hispanic (57.1%; *n* = 105), gay-identifying (81.2%; *n* = 151), and cisgender men (88.7%; *n* = 165). One in five participants completed high school or lower (19.4%; *n* = 36). Among those who had never used PrEP (53.4%; *n* = 101), nearly all participants were willing to use it if it were free or covered as part of their insurance (89.4%; *n* = 84). The most common barriers to not using PrEP was not knowing where to obtain it (68.3%; *n* = 69), concerns about side effects (42.1%; *n* = 35), and concerns about affordability (38.5%; *n* = 25)**.** This study discusses specific nuances to the suburbs, including cultural norms and structural barriers that should be incorporated in health promotion initiatives in addressing these factors.

## 1. Introduction

In the United States (U.S.), the HIV epidemic continues to disproportionately impact gay, bisexual, and other men who have sex with men (MSM) and transgender women. The U.S. Centers for Disease Control and Prevention (CDC) reported half of all people living with HIV (PLWH) are gay, bisexual, or other MSM [1]. In 2019, 69% of new HIV diagnoses in the country were categorized as MSM transmission [2]. The epidemic has had a particular deleterious impact on gay and bisexual men of color, especially Black/African Americans and Latinx/Hispanics. Amongst MSM, Black/African Americans constitute 42% of all new HIV diagnoses, and Latinx/Hispanics comprise 21% [1], whereas Black/African Americans and Latinx/Hispanic people make up 13% and 18% of the entire population, respectively [3,4]. Based on rates of diagnosis, the CDC estimates that gay and bisexual men will continue to be the most affected by HIV in the U.S., with half of Black/African American gay and bisexual men and one-quarter of Latinx/Hispanic gay and bisexual men acquiring HIV in their lifetimes, thus showing the importance of assessing new prevention technologies and their uptake in the community [5].

The HIV epidemic in Long Island, the suburbs east of New York City, is the largest suburban epidemic in the U.S. with over 5700 diagnosed PLWH and a prevalence rate of 195 per 100,000 people [6]. Excluding prisoners, the largest transmission category of PLWH on Long Island are MSM [6]. Racially speaking, non-Hispanic White MSM constituted 31% of all PLWH on Long Island, Latinx/Hispanic MSM comprised 29%, and non-Hispanic Black MSM comprised 24% [6]. Amongst new diagnoses in 2016, 49% were MSM [6]. Advances are needed to increase the uptake of HIV prevention technologies, such as pre-exposure prophylaxis (PrEP) on Long Island, especially for MSM and especially for Black/African American and Latinx/Hispanic MSM.

PrEP usage remains inconsistent in the U.S. amongst MSM. Various factors affect PrEP uptake; within the racially and socioeconomically diverse gay communities, cultural norms about taking PrEP have differed widely. Black/African American and Latinx/Hispanic MSM are less likely than their White counterparts to talk with a medical professional about PrEP [7]. Black/African American MSM have demonstrated higher rates of stigma about PrEP usage than White counterparts [8]. These social networks become important in terms of validating or stigmatizing behaviors. In some cases, stigma against PrEP use is rooted firmly in the perception that all PrEP users are promiscuous [8,9,10]. Knowledge about PrEP is an integral part of MSM acceptance of PrEP use [11]; however, PrEP awareness amongst MSM varies depending on racial/ethnic group. In the U.S., social marketing campaigns continue to target Black/African American and Latinx/Hispanic MSM [7,12].

Public health interventions in the U.S. have focused on the unique challenges of urban and rural populations. While urban conditions include dense populations and multiple points of contact, rural health is characterized by distance between people and healthcare opportunities [13]. Suburbs have demonstrated a different matrix: often increased economic capital and a decrease in social capital as people make more money per capita, but connect with each other less [14]. Long Island has demonstrated these dynamics, particularly with those ethnic groups who could become categorized as White in the U.S. Where once mutual aid societies and civic associations were a vital part of Long Island life, they have since diminished as generations become acclimated to suburban norms [15]. This complicates public health education interventions that rely on social networks to transmit information.

There is a dearth of research on PrEP uptake, social support, and personal attitudes on MSM within a suburban context [16]. As more and more state and regional HIV plans prioritize PrEP usage for HIV-negative MSM, we need to have a fuller view of how PrEP can be incorporated in suburban settings including developing community education tools and interventions. Thus, this exploratory study examines attitudes and experiences with PrEP amongst gay and bisexual men and transgender women in suburban Long Island. Challenges and facilitators to PrEP uptake must be explored on Long Island in order to increase PrEP uptake across the State of New York, and not just within the domains of New York City.

## 2. Materials and Methods

Long Island is a complex suburban landscape. It is historically racially segregated, with a growing Latinx/Hispanic population [17]. The Long Island Rail Road and the Metropolitan Transit Authority provide ready access to New York City, with the largest urban HIV epidemic in the country [18]. Long Island is split between Suffolk County and Nassau County, but the geography of the island includes Kings and Queens Counties (Brooklyn and Queens, respectively) on its landmass. As Kings and Queens Counties are part of New York City, the current study focuses on Suffolk and Nassau specifically. These two counties differ widely in terms of density and race. Nassau is more populated and racially diverse: Suffolk has more rural areas too, and it was once mostly rural. Since the 1970s, this has changed as the suburbs surrounding New York City grew with both flight from the city and people moving into the area for work [19]. As a large suburban area, Long Island has a diverse range of communities and cultures. Some enjoy their access to the most populous city in the US; others are content to only explore Long Island.

We implemented an online cross-sectional survey of MSM and transgender women. The survey was administered in 2017 to eligible individuals 18 or older living on Long Island. The convenience sample was recruited via medical and social service providers, community-based organizations (CBOs), universities, social media, and word-of-mouth. The study was approved by the university Institutional Review Board (approval no. 20170605-HPHS-SAN-1). Consent was obtained via the survey from all participants prior to survey administration.

The survey included 65 questions about PrEP awareness, PrEP use, reasons for intentions to use PrEP, sexual behaviors in the last 12 months, dating characteristics, and sociodemographic characteristics. Demographic (age, education, race, ethnicity, born in the U.S., zip code, sexual orientation, insurance status, annual income, marital status, and gender identity), sexual behavior/sexual partner history (number of sex partners, sex partners HIV status, venue of meeting sex partners, type of sexual behaviors, HIV/STI testing history, condom use during sex), STI history (testing of participant and their sex partners, frequency of testing), and drug use (drug use during sex) questions were adapted from the Seattle Pride Survey [20]. Questions regarding reasons for intentions to use PrEP were adapted from prior studies of nPEP conducted at Fenway Community Health in Boston (knowledge of PrEP, frequency and mode of PrEP use, sex behaviors as a result of PrEP use, PrEP use of friends and sex partners) [21].

We present descriptive findings of having used PrEP, and bivariate analyses by sociodemographic and sexual behavior reports, as well as knowledge of and attitudes toward PrEP. Furthermore, we used these measures to estimate the HIV incidence risk index for men who have sex with men (HIRI-MSM), a tool used by clinicians to identify patients to target for PrEP discussions [22]. This score was used to create very low, low, high and very high risk categories, as well as dichotomized to high versus low risk. We calculated chi-squared tests, or Fisher’s exact tests when cell sizes were small, to compare categorical variables, and *t*-tests for continuous outcomes. We also ran a modified Poisson regression with robust standard error estimates in a multivariable model exploring predictors of reporting having used PrEP.

## 3. Results

A total of 262 individuals provided consent and began the questionnaire. Individuals who did not complete the questionnaire (*n* = 28), reported living with HIV (*n* = 13), reported living outside of Long Island (*n* = 20), or did not identify as a cisgender man or transgender woman (*n* = 12) were excluded, resulting in an analytic sample of 189 participants. Of these 189 participants, six participants are excluded from analyses on PreP use as they did not respond to this question or indicated ‘do not know’ as their response.

Sample demographic and behavioral characteristics are displayed in Table 1. The mean age in the sample was 28.7 years (Standard Deviation [SD]: 12.0, range 18–68). Of the 183 participants who responded to the PrEP use question, most participants were Latinx/Hispanic (56.6%; *n* = 107), gay-identifying (82.0%; *n* = 155), and cisgender men (88.4%; *n* = 167). One in five participants completed high school or lower (19.6%; *n* = 37). Nearly all participants had some form of health insurance (91.0%; *n* = 172), with one-third having private health insurance (32.6%; *n* = 56). Two-thirds of participants had tested for HIV in the past 12 months (67.7%; *n* = 128). Participants reported a median of three partners (Interquartile Range [IQR]: 1–5) in the last 90 days. Two-thirds (68.8%; *n* = 130) reported use of alcohol or other drugs during sex in the last 90 days.

In this sample, 44.4% (*n* = 84) had used PrEP. Bivariate analyses assessing differences in PrEP use by demographic and behavioral characteristics are displayed in Table 1. Individuals who had used PrEP were older and were more commonly gay-identifying (versus non-gay-identifying), cisgender men (versus transwomen), Latinx/Hispanic (versus non-Latinx/Hispanic), completed at least some college (versus completing high school or less), reported an annual income of USD 50,001 or higher (versus USD 50,000 or lower), had private insurance (versus public insurance or no insurance coverage), had been tested for HIV in the past 12 months (versus having been tested for HIV more than 12 months ago), had more sex partners in a 12 month timeframe, and had used alcohol or other drugs during sex (versus not using drugs during sex) (all *p* < 0.01).

Bivariate analyses assessing differences in PrEP use by personal experiences with PrEP are displayed in Table 2. Those who considered themselves at higher risk of HIV acquisition were more likely to say they had used PrEP than those who considered themselves at low HIV risk. PrEP usage amongst social and sexual networks also seemed to play a role, as those with friends or sex partners who had taken PrEP were more likely to have used PrEP than those whose friends or sex partners had not taken it (all *p* < 0.01).

The bivariate analyses assessing differences in PrEP use by questions about willingness to take PrEP are listed in Table 3. People were less likely to have used PrEP in their lives if they were unwilling to take PrEP in any way beyond the single pill, once daily regimen. In addition, concerns about costs showed up amongst people who had never used PrEP in their lifetime. Amongst respondents who had never used PrEP in their lifetime, 100% indicated they would take PrEP if their sex partner was living with HIV.

The results of multivariable analyses identifying correlates of PrEP use are shown in Table 4. Independent of other characteristics, Latinx/Hispanic individuals were nearly six times more likely to have used PrEP (adjusted prevalence ratio, aPR: 5.83; 95% CI: 1.74–2.29) and those with annual income of USD 50,000 or less were 80% less likely to have used PrEP (aPR: 0.20; 95% CI: 0.05–0.57). Among those who had never used PrEP (*n* = 105), nearly all participants were willing to use it if it were free or covered as part of their insurance (83.8%; *n* = 88). The most common barriers to not using PrEP were concerns about side effects (34.3%; *n* = 36), not knowing where to obtain it (30.5%; *n* = 32), and concerns about affordability (24.8%; *n* = 28)**.**

## 4. Discussion

Statistical analyses revealed several factors that indicated facilitation or acted as a barrier to research participants’ knowledge of and willingness to use PrEP. The supportive factors include social and sexual networks, as well as individual identity. Barriers include cost, access, and beliefs about the efficacy of the medication regimen.

PrEP use amongst social and sexual networks was correlated with personal history of PrEP usage. When a person’s friends and sexual partners take PrEP, there is a high likelihood that the social norm in the group allows for PrEP as part of HIV prevention strategies. In addition, the knowledge that people are taking PrEP in a social and sexual network further supports the notion that this is a social norm, as people disclose and discuss PrEP use [23]. By making PrEP use normative and acceptable, these social and sexual networks further support the uptake of this HIV prevention technology [24]. Increasing social and community norms about the use of HIV prevention technologies such as PrEP could greatly increase uptake [23].

Social and sexual networks have had a positive impact on other HIV prevention strategies. When gay and bisexual men are more closely identified with the gay community, they are more likely to read, understand and feel implicated in HIV prevention messaging aimed at the gay community [25]. For PrEP, researchers found that the absence of stigma around PrEP in a social network facilitated PrEP retention for gay and bisexual men in a Mississippi clinic [9]. There could be more interventions targeting social norms around PrEP, decreasing stigma around PrEP use (including anti-sex messages), and increasing conversations amongst social networks around HIV prevention technologies.

While this study did not specifically look at social networks, there were some findings that were better understood in light of how these networks function to disseminate information. Latinx/Hispanic identification had a higher correlation with PrEP knowledge and willingness to use it. Other research has found that Latinx/Hispanic MSM tended to have lower knowledge about PrEP [26,27] so this is a promising data point. Latinx/Hispanic MSM might be more willing to use PrEP for a variety of reasons, including social marketing as Long Island has had some State health department funded initiatives meant to educate the MSM community about PrEP in the past few years. In addition, many of the research participants were recruited from lesbian, gay, bisexual, transgender, and queer/questioning (LGBTQ+) and HIV CBOs and this involvement with CBOs could also be associated with more knowledge of PrEP and more exposure to HIV prevention messages [25]. Finally, Vega et al. [28] found that Latinx/Hispanic MSM in Queens, New York built rich social connections to share information about a wide range of issues, including sex venues, friendly health service organizations, immigration challenges, and treatment options. These connections could have facilitated PrEP knowledge. Further research in this area is needed.

Some barriers were also identified by research participants, including cost and access. If people could obtain PrEP for free, they were more likely to consider using it. This supports other studies on PrEP barriers where cost is identified as a key barrier [29,30,31]. In order to encourage PrEP uptake, attention must be paid to cost. When free or low-cost PrEP is available, this should be a feature in social marketing and health education initiatives so that potential users are aware of it.

Another barrier to uptake was how regularly PrEP should be taken. Research participants indicated they were less likely to use PrEP if it was not one pill daily (regardless of whether it was more or less than once daily). More than one pill could be considered a nuisance and too much work for the result. Since much of the messaging on PrEP is a once-daily medication, the MSM community may not trust any other variation from this as effective. This has particular concern as different PrEP regimens are tested. While it may seem as though on-demand PrEP (where individuals take PrEP at certain time intervals based on intent and need) will be effective with individuals who have difficulty taking one pill daily, there would need to be extensive community education to build trust in this new regimen.

## 5. Conclusions

PrEP is an integral tool to HIV prevention efforts across the country. In every jurisdiction, large scale “Ending the HIV Epidemic” plans are being developed that include PrEP yet there is little research on PrEP in non-urban settings. While PrEP is a useful technology, it is clear that uptake is irregular amongst gay and bisexual men. This study points to social and sexual networks, as well as anti-stigma work, as productive points of intervention. If HIV prevention plans include large-scale uptake of PrEP, several implementation factors have to be included, such as: mechanisms to build norms in social and sexual networks that facilitate discussion of PrEP and other HIV prevention technologies, work against stigma against PrEP including stigma against gay sex, and communication campaigns to update communities as scientific advances allow for greater variety of PrEP medications and regimens.

This study expands the knowledge about suburban health, particularly for gay and bisexual men. Whereas most studies of gay and bisexual men focus on urban centers, this explores gay and bisexual men’s health on Long Island. Suburban health studies rarely include LGBTQ+ communities [16]. This exclusion is problematic as it can limit how LGBTQ+-sensitive health services will be delivered in suburban contexts, as well as what kinds of health issues are considered relevant. There are specific nuances to regions, such as ways of communicating information, cultural norms, access to transportation, socio-economic and racial divides, and these impact the ways that health interventions are researched, designed, and implemented.

## Figures and Tables

**Table 1 ijerph-19-11640-t001:** Demographic characteristics of a community-based sample of cisgender men and transgender women who have sex with men residing on Long Island, stratified by lifetime history of pre-exposure prophylaxis (PrEP) use, 2017.

	Total (*n* = 183) *	Lifetime History of PrEP Use		*p* Value ^
		No (*n* = 99)	Yes (*n* = 84)	
**Mean Age in years (SD)**	28.2 (11.0)	29.8 (14.5)	26.3 (3.8)	0.03
**Race/Ethnicity % (*n*) ****				<0.01
White (non-Hispanic/Latino)	34.3 (62)	93.5 (58)	6.5 (4)	
Black/African American (non-Hispanic/Latino)	5.0 (9)	77.8 (7)	22.2 (2)	
Hispanic/Latino	58.0 (105)	26.7 (28)	73.3 (77)	
Other	2.8 (5)	100.0 (5)	--	
**County of Residence % (*n*) ****				0.07
Suffolk	50.0 (91)	47.3 (43)	52.7 (48)	
Nassau	50.0 (91)	61.5 (56)	38.5 (35)	
**Sexual Identity % (*n*) ****				<0.01
Men who have sex with men	90.7 (165)	50.3 (83)	49.7 (82)	
Transgender women	9.3 (17)	94.1 (16)	5.9 (1)	
**Educational Attainment % (*n*) ****				<0.01
High school graduate or less	19.8 (36)	80.6 (29)	19.4 (7)	
Some college, two-year degree, or technical school	55.5 (101)	35.6 (36)	64.4 (65)	
College graduate or higher	24.7 (45)	75.6 (34)	24.4 (11)	
**Annual Income % (*n*) ****				<0.01
No income	4.0 (7)	85.7 (6)	14.3 (1)	
USD 1 to USD 25,000	27.3 (48)	93.7 (3)	6.3 (3)	
USD 25,001 to USD 50,000	12.5 (22)	95.5 (21)	4.5 (1)	
USD 50,001 to USD 75,000	47.2 (83)	20.5 (17)	79.5 (66)	
USD 75,001 or greater	9.1 (16)	31.2 (5)	68.8 (11)	
**Health Insurance Status % (*n*) ****				<0.01
No health insurance	4.7 (8)	87.5 (7)	12.5 (1)	
Private health insurance	31.6 (55)	90.9 (50)	9.1 (5)	
Medicaid	48.9 (85)	22.4 (19)	77.6 (66)	
Medicare	11.5 (20)	45.0 (9)	55.0 (11)	
Other	3.4 (6)	100.0 (6)	--	
**Diagnosed with Depression % (*n*) ****	55.3 (99)	331.3 (31)	68.7 (68)	<0.01

* excludes six participants who reported ‘don’t know’ or did not respond to the question on PrEP use. ** sample size may vary due to missing responses. ^ Chi-squared or Fisher’s exact test for categorical variables and Kruskal–Wallis h-test comparing use of PreP by characteristics.

**Table 2 ijerph-19-11640-t002:** Behavioral characteristics of a community-based sample of cisgender men and transgender women who have sex with men residing on Long Island, stratified by lifetime history of pre-exposure prophylaxis (PrEP) use, 2017.

% (*n*) **	Total (*n* = 183) *	Lifetime History of PrEP Use		*p* Value ^
		No (*n* = 99)	Yes (*n* = 84)	
**Risk for HIV Infection** **(HIRI-MSM Index)**				<0.001
Very low risk for HIV infection	13.2 (24)	95.8 (23)	4.2 (1)	
Low risk for HIV infection	6.6 (12)	91.7 (11)	8.3 (1)	
Elevated risk for HIV infection	48.4 (88)	51.1 (45)	48.9 (43)	
Highest risk for HIV infection	31.9 (58)	34.5 (20)	65.5 (38)	
**Has a friend who has taken PrEP**				<0.01
Yes	75.3 (137)	40.9 (56)	59.1 (81)	
No	15.4 (28)	96.4 (27)	3.6 (1)	
Don’t know	9.3 (21)	94.1 (16)	5.9 (1)	
**Has a sex partner who has taken PrEP**				<0.01
Yes	69.8 (127)	36.2 (46)	63.8 (81)	
No	18.7 (34)	97.1 (33)	2.9 (1)	
Don’t know	11.5 (21)	95.2 (20)	4.8 (1)	
**Ever diagnosed with an STI**	62.2 (112)	31.2 (35)	68.8 (77)	<0.01
**Frequency of STI testing**				<0.01
Never	11.1 (20)	95.0 (19)	5.0 (1)	
Every few years or once per year	45.3 (82)	40.2 (33)	59.8 (49)	
About every 6 months	23.2 (42)	40.5 (17)	59.5 (25)	
About every 3 months	20.4 (37)	78.4 (29)	21.6 (8)	
**Engaged in receptive anal intercourse**	82.8 (149)	46.3 (69)	53.7 (80)	<0.01
**Frequency of receptive anal intercourse**				<0.01
Never	23.6 (43)	97.7 (42)	2.3 (1)	
Most of the time	27.5 (50)	26.0 (13)	74.0 (37)	
Sometimes	27.5 (50)	16.0 (8)	84.0 (42)	
Once or a few times	11.0 (20)	95.0 (19)	5.0 (1)	
All of the time	9.9 (18)	88.9 (16)	11.1 (2)	
**Engaged in insertive anal intercourse**	81.7 (147)	45.6 (67)	54.4 (80)	<0.01
**Frequency of insertive anal intercourse**				<0.01
Never	19.0 (34)	94.1 (32)	5.9 (2)	
Most of the time	46.9 (84)	17.9 (15)	82.1 (69)	
Sometimes	10.6 (19)	52.6 (10)	47.4 (9)	
Once or a few times	12.9 (23)	95.7 (22)	4.3 (1)	
All of the time	10.6 (19)	89.5 (17)	10.5 (2)	
**HIV Status of Sexual Partners**				
Had at least one HIV-positive partner	6.9 (12)	50.0 (6)	50.0 (6)	0.87
Had at least one HIV-negative partner and had no reason to doubt their status	70.8 (126)	35.7 (45)	64.3 (81)	<0.01
Had at least one unknown status partner or HIV-negative partner with reason to doubt their status	67.6 (121)	35.5 (43)	64.5 (78)	< 0.01
**Drugs Used During Sex**				
Poppers	17.6 (32)	87.5 (28)	12.5 (4)	< 0.01
MDMA	0.6 (1)	100.0 (1)	--	0.54
Marijuana	26.9 (49)	59.2 (29)	40.8 (20)	0.43
Crystal methamphetamine	0.6 (1)	--	100.0 (1)	0.46
Alcohol	56.6 (103)	38.8 (40)	61.2 (63)	< 0.01

HIRI-MSM = HIV Incidence Risk Index for Men who have Sex with Men; STI = sexually transmitted infection; MDMA = * excludes six participants who reported ‘don’t know’ or did not respond to the question on PrEP use. ** sample size may vary due to missing responses. ^ Chi-squared or Fisher’s exact test comparing characteristics by lifetime history of PrEP use.

**Table 3 ijerph-19-11640-t003:** Willingness to use pre-exposure prophylaxis among a community-based sample of cisgender men and transgender women who have sex with men residing on Long Island, 2017.

% (n)	Total (*n* = 183) *	Lifetime History of PrEP Use		*p* Value ^
		No (*n* = 99)	Yes (*n* = 84)	
More likely to use PrEP every day if thought it worked	74.2 (135)	40.7 (55)	59.3 (80)	<0.01
Willing to take PrEP before hot date, but only a single dose	6.0 (11)	90.9 (10)	9.1 (1)	0.01
Willing to take PrEP before hot date and daily for 28 days after	7.1 (13)	92.3 (12)	7.7 (1)	<0.01
Willing to use PrEP for all condomless anal intercourse	30.8 (56)	64.3 (36)	35.7 (20)	0.07
Not willing to take PrEP if it meant more than one pill	5.5 (10)	90.0 (9)	10.0 (1)	0.02
Not willing to take PrEP if it had to be taken more than once per day	5.5 (10)	90.0 (9)	10.0 (1)	0.02
Willing to only use PrEP if partner was HIV-infected	6.6 (12)	100.0 (12)	--	<0.01
Not willing to take PrEP due to concerns about side effects	21.8 (36)	91.7 (33)	8.3 (3)	<0.01
Know how to obtain PrEP if want it	82.3 (151)	45.0 (68)	55.0 (83)	<0.01
Cannot afford PrEP	18.9 (28)	89.3 (25)	10.7 (3)	<0.01
More inclined to use PrEP if free or covered by insurance	94.9 (167)	60.3 (84)	49.7 (83)	<0.01

* excludes six participants who reported ‘don’t know’ or did not respond to the question on PrEP use. ^ Chi-squared or Fisher’s exact test comparing characteristics by lifetime history of PrEP use.

**Table 4 ijerph-19-11640-t004:** Multivariable analyses of correlates of lifetime PrEP use among a community-based sample of cisgender men and transgender women who have sex with men residing on Long Island, 2017.

	Unadjusted Model PR (95% CI)	Adjusted Model aPR (95% CI)
**Race/Ethnicity**		
White (Non-Hispanic/Latinx)	Referent	Referent
Other Race	10.3 (3.41, 30.9)	3.60 (0.19, 68.3)
**Education Level**		
Some college or higher	Referent	Referent
High school graduate or less	0.37 (0.19, 0.73)	0.73 (0.45, 1.17)
**Health Insurance Coverage**		
Public/Private Insurance	Referent	Referent
No Health Insurance	0.22 (0.04, 1.41)	1.52 (0.20, 11.9)
**History of Depression**		
No	Referent	Referent
Yes	3.64 (2.25, 5.91)	1.54 (1.08, 2.21)
**HIRI-MSM Score**		
Low risk	Referent	Referent
High risk	12.5 (1.83, 85.6)	2.72 (0.33, 22.7)
**Concern about Side Effects**		
No	Referent	Referent
Yes	0.14 (0.05, 0.42)	0.82 (0.21, 3.21)
**Concern about Affordability**		
No	Referent	Referent
Yes	0.18 (0.06, 0.52)	0.32 (0.09, 1.19)

PR = Prevalence ratio, aPR = adjusted prevalence ratio, HIRI-MSM = HIV incidence risk index for men who have sex with men.

## Data Availability

Not applicable.

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
