# Peer review of "Knowledge, Attitudes, and Practices towards PrEP from Cisgender Men and Transgender Women Who Have Sex with Men in the Largest Suburban HIV Epidemic"

_ijerph, 2022, doi:10.3390/ijerph191811640_

Round 1

Reviewer 1 Report

The authors are treating an important subject in the prevention of HIV in all world populations, the US and foreign (This review comes from Brazil). They have generally treated it adequately and provided interesting results. However, the study still needs work to bring it to the high level of quality that will make it useful to clinicians and the PLHV communities around the world.

1. The English is generally clear. However, there are some instances of word usage that are not acceptable. Particularly, the word "ever" being used as the opposite of "never" is very forced and initially was difficult to understand. It may be acceptable in specific academic communities in the US, but outside, not at all. It's a neologism that is attempting to replace common English usage. Stick to simple, clear English.

2. In the Introduction, you describe 3 social dynamics for Long Island (I grew up in Nassau County), which were interesting. Then you drop the three and just describe the suburban dynamic. If you are going to raise the issue of the three dynamics, then you need to describe all three and how they interact. For Nassau and Suffolk, I really saw 2 -- suburban and rural-- when I grew up and when I lived there in the 70's and 80's. Focus on those and particularly how the rural/agricultural has been transformed into suburban. On this point, I found the Hamptons reference gratuitous and unnecessary. It leaves out other second-home communities like Fire Island, etc., etc. 

3. You had 81% of your sample identifying as gay, which left 38 respondents as not identifying as gay. I would be very reticent to draw any conclusions about that group in your results because it is small and subject to a lot of variance. As well, it may not be a term that means much since it certainly includes unacknowledged gay men as well as possibly other groups.

4. The p-values in Tables 1 - 3 mystified me. What differences did they test? The never vs. ever columns or the sub-categories of each of the headed groups. They left me confused. Given that this is a survey with 65 questions and 189 responses, I think you are on stronger statistical ground to use CI's instead of p-values as your measure.

5. Your discussion focuses on the importance of social networks, but your study doesn't really address that issue, which I agree is super important to spread the knowledge through the community and give the use of Prep some oomph. How do your results strengthen or weaken social networks may be a good way to integrate it better into this study.

Author Response

The authors are treating an important subject in the prevention of HIV in all world populations, the US and foreign (This review comes from Brazil). They have generally treated it adequately and provided interesting results. However, the study still needs work to bring it to the high level of quality that will make it useful to clinicians and the PLHV communities around the world.

Thank you.

1. The English is generally clear. However, there are some instances of word usage that are not acceptable. Particularly, the word "ever" being used as the opposite of "never" is very forced and initially was difficult to understand. It may be acceptable in specific academic communities in the US, but outside, not at all. It's a neologism that is attempting to replace common English usage. Stick to simple, clear English.

Thank you for this comment.  We have changed the language to “have used Prep” or “lifetime history of Prep use” rather than “have ever used prep,” to make the language clearer throughout the revised manuscript.

  1. In the Introduction, you describe 3 social dynamics for Long Island (I grew up in Nassau County), which were interesting. Then you drop the three and just describe the suburban dynamic. If you are going to raise the issue of the three dynamics, then you need to describe all three and how they interact. For Nassau and Suffolk, I really saw 2 -- suburban and rural-- when I grew up and when I lived there in the 70's and 80's. Focus on those and particularly how the rural/agricultural has been transformed into suburban. On this point, I found the Hamptons reference gratuitous and unnecessary. It leaves out other second-home communities like Fire Island, etc., etc.

Thank you for your input. We were attempting to show the broad economic divide on the island. We have adjusted the text discussing both the rural and suburban aspects of Long Island populations.

  1. You had 81% of your sample identifying as gay, which left 38 respondents as not identifying as gay. I would be very reticent to draw any conclusions about that group in your results because it is small and subject to a lot of variance. As well, it may not be a term that means much since it certainly includes unacknowledged gay men as well as possibly other groups.

Thank you for this comment.  81.2% were gay-identifying as noted in the abstract – note that our population included bisexual individuals and transgender women.  We are careful not to make conclusions about transgender women, and do not run separate analyses for gay versus bi-sexual identifying MSM, given concerns around sample size, but do present findings from all participants who met our eligibility criteria.

  1. The p-values in Tables 1 - 3 mystified me. What differences did they test? The never vs. ever columns or the sub-categories of each of the headed groups. They left me confused. Given that this is a survey with 65 questions and 189 responses, I think you are on stronger statistical ground to use CI's instead of p-values as your measure.

Thank you for this suggestion.  We have clarified the tests used to generate the p-values in Tables 1-3 (in addition to describing these in the methods section).  We prefer to include the p-values in bivariate analyses, and present 95% confidence intervals for the multivariable model (Table 4).

  1. Your discussion focuses on the importance of social networks, but your study doesn't really address that issue, which I agree is super important to spread the knowledge through the community and give the use of Prep some oomph. How do your results strengthen or weaken social networks may be a good way to integrate it better into this study.

 Social networks are previewed in the start of the article, and then followed up in the discussion. We have added some language to frame its use and relationship to this study.

The manuscript by Spieldenner et al analyzed the data from a cross-sectional web-based study to figure out what factors could affect the promotion of HIV PrEP which was very important for HIV prevention. Unlike other studies which focused attention on urban region, this study explored the attitudes and experiences of gay, bisexual, other men who have sex with men (MSM) and transgender women on PrEP in suburban Long Island. They found that social and sexual networks had a positive impact on the PrEP promotion and HIV prevention in this suburban region. People whose friends or sexual partners had taken PrEP and gay-identifying participants were more willing to accept PrEP, but excessive cost, inaccessibility and frequency of use could put barrier for people to use PrEP. Overall, this paper could give us some useful information to make plans to promote PrEP in suburb. However, there are some questions remaining to be addressed, especially data accuracy. Please see the following comments.

Major points

1.  Page 3, line 123, 262-28-15-20-12=187, why the authors described as 189 participants? Maybe should be 187. If the number 189 was right, why the total number of participants was 182 in every table?

189 is the full sample. 183 is the number who provided an answer to the PrEP use history (6 participants selected don't know or prefer not to answer and are excluded from the lifetime history of use yes/no dichotomy).  We have added a footnote to the tables to make this clear.

  1. Some data shown in the main body were different from these in the tables. For example, page 3, line 126-134, Latinx (57.1%; n = 105), cisgender men (88.7%; n = 165), health insurance (95.9%; n = 167), 43.3% (n = 82) had ever used PrEP and so on, these data didn’t match these in table 1. I just listed some of the mistakes, not all. Please check carefully to figure out all the mistakes and correct them.

Thank you for this comment.  In the opening paragraph of results, we originally presented statistics on the full sample of 189.  However, Table 1 presents data for the reduced sample of 183, excluding the 6 don’t know/no response to the lifetime use of PrEP.  As noted above, we have indicated this more carefully in the manuscript and have double-checked all of the remaining estimates in the text.

  1. Some data shown in the main body could not be found in the tables. For example, page 3, line 126-134, gay-identifying (81.2%; n = 151), two-thirds of participants had tested for HIV in the past 12 months (69.6%; n = 128), Nearly three-quarters (70.4%; n = 131) and so on. These data couldn’t be found in table 1 or any one table. It would be better to show all the useful data in the tables.

We have included the data in Table 1 that we considered most important for bivariate and multivariable results presented in Tables, 2-4, but have included other contextual information in the text that is not included in the Tables to avoid redundancy in findings.

  1. Table 1, why the total number of participants listed under Race/Ethnicity, Annual Income and Health Insurance Status didn’t match those numbers (182, 99, 83) shown in the first line of this table. For example, the total number listed under Race/Ethnicity was 62+9+105+5=181, not 182. Table 2 had the same mistakes as table 1. Please check carefully to figure out all the mistakes and correct them.

Thank you for this question – some measures are missing responses so not all counts sum to 183; we have clarified this by adding an asterisk to the table that sample sizes vary due to missing responses. 

  1. Table 3, many data in this table were wrong. If they are percentages, why 55/99=40.7%, 80/83=59.3%...? And as shown in this table, the percentage of “Know how to get PrEP if want it” is 82.3 (151). Why the percentage of “not knowing where to get it” is 68.3% (n = 69) as shown on page1 line 23, page 4 line 163-164. It didn’t make sense.

Thanks for this comment.  We are presenting row percentages (e.g., 55 out of 135) rather than column percentages (e.g., 55 out of 99) for the lifetime history estimates (sum to 100%).

  1. Table 4 was unreadable to readers. Please make it easier.

Thank you for this comment.  We have updated the title of the table to make it clearer that we are presenting crude and adjusted prevalence ratios for having used PreP during the respondent’s lifetime by socio-demographic characteristics.

Minor points

1.  Table 1, it would be better to show both age range and average age.

Thank you for this suggestion.  We have reported the mean age and age range of the participants in the text (Line 204), rather than included it in the Table.

Thank you for these comments, they have helped to strengthen the manuscript.  We also updated our language for ‘Latinx’ to be ‘Latinx/Hispanic’ given recent data to suggest that most people who identify as Hispanic to not understand or identify with the term Latinx.

Reviewer 2 Report

The manuscript by Spieldenner et al analyzed the data from a cross-sectional web-based study to figure out what factors could affect the promotion of HIV PrEP which was very important for HIV prevention. Unlike other studies which focused attention on urban region, this study explored the attitudes and experiences of gay, bisexual, other men who have sex with men (MSM) and transgender women on PrEP in suburban Long Island. They found that social and sexual networks had a positive impact on the PrEP promotion and HIV prevention in this suburban region. People whose friends or sexual partners had taken PrEP and gay-identifying participants were more willing to accept PrEP, but excessive cost, inaccessibility and frequency of use could put barrier for people to use PrEP. Overall, this paper could give us some useful information to make plans to promote PrEP in suburb. However, there are some questions remaining to be addressed, especially data accuracy. Please see the following comments.

Major points

11.       Page 3, line 123, 262-28-15-20-12=187, why the authors described as 189 participants? Maybe should be 187. If the number 189 was right, why the total number of participants was 182 in every table?

22.       Some data shown in the main body were different from these in the tables. For example, page 3, line 126-134, Latinx (57.1%; n = 105), cisgender men (88.7%; n = 165), health insurance (95.9%; n = 167), 43.3% (n = 82) had ever used PrEP and so on, these data didn’t match these in table 1. I just listed some of the mistakes, not all. Please check carefully to figure out all the mistakes and correct them.

33.       Some data shown in the main body could not be found in the tables. For example, page 3, line 126-134, gay-identifying (81.2%; n = 151), two-thirds of participants had tested for HIV in the past 12 months (69.6%; n = 128), Nearly three-quarters (70.4%; n = 131) and so on. These data couldn’t be found in table 1 or any one table. It would be better to show all the useful data in the tables.

44.       Table 1, why the total number of participants listed under Race/Ethnicity, Annual Income and Health Insurance Status didn’t match those numbers (182, 99, 83) shown in the first line of this table. For example, the total number listed under Race/Ethnicity was 62+9+105+5=181, not 182. Tale 2 had the same mistakes as table 1. Please check carefully to figure out all the mistakes and correct them.

55.       Table 3, many data in this table were wrong. If they are percentages, why 55/99=40.7%, 80/83=59.3%...? And as shown in this table, the percentage of “Know how to get PrEP if want it” is 82.3 (151). Why the percentage of “not knowing where to get it” is 68.3% (n = 69) as shown on page1 line 23, page 4 line 163-164. It didn’t make sense.

66.       Table 4 was unreadable to readers. Please make it easier.

Minor points

11.       Table 1, it would be better to show both age range and average age.

Author Response

(The authors gave the same response as above.)

Round 2

Reviewer 2 Report

The authors have addressed my concerns. I don't have any other questions.